# Role of Thyroid Hormone in Neurodegenerative Disorders of Older People

**DOI:** 10.3390/cells14020140

**Published:** 2025-01-18

**Authors:** Arshag D. Mooradian, Michael J. Haas

**Affiliations:** Division of Endocrinology, Diabetes, and Metabolism, Department of Medicine, University of Florida Jacksonville College of Medicine, 653-1 West 8th Street, Jacksonville, FL 32209, USA; michael.haas@jax.ufl.edu

**Keywords:** neurodegeneration, thyroid hormones, Parkinson’s disease, Alzheimer’s disease

## Abstract

Thyroid dysfunction is associated with a number of neuropsychiatric manifestations. Cognitive decline is a common feature of hypothyroidism and clinical or subclinical hyperthyroidism. In addition, there is a significant association between thyroid hormone (TH) levels and the degree of cognitive impairment in Parkinson’s disease (PD). The pathophysiology of TH-related neurodegeneration include changes in the blood–brain barrier, increased cellular stress, altered processing of β-amyloid precursor protein and the effect of TH on neuronal cell viability. The neurotoxicity of TH is partially mediated by the thyroid hormone responsive protein (THRP). This protein is 83% homologous to mouse c-Abl-interacting protein-2 (Abi2), a c-Abl-modulating protein with tumor suppressor activity. In cell cultures, increasing THRP expression either with TH treatment or exogenously through transfecting neuronal or PC 12 cells causes cell necrosis. The expression of exogenous THRP in other cells such as the colonic epithelial cell line Caco-2 and the glial cell line U251 has no effect on cell viability. The effect of THRP on cell viability is not modulated by c-Abl tyrosine kinase. The causal relationship between specific biochemical perturbations in cerebral tissue and thyroid dysfunction remains to be elucidated.

## 1. Introduction

Thyroid dysfunction in adults is associated with a number of neuropsychiatric manifestations [1,2,3,4,5,6]. The precise underlying mechanism of the changes in the central nervous system (CNS) is not known. Abnormal blood levels of thyroid hormones (THs) affect neurotransmission, cause oxidative and inflammatory stress, alter the blood–brain barrier (BBB) function, and modulate gene expression in neuronal and glial cells [1]. The neuropsychiatric manifestations of thyroid disease are largely reversible with the normalization of blood concentrations of TH. However, the roles of THs in neurodegenerative disorders of older people such as Alzheimer’s disease (AD) and Parkinson’s disease (PD) have not been well studied. Over the last two decades, evidence has emerged as to the effect of THs on neuronal cell viability that may contribute to the emergence of neurodegenerative disorders. In this communication, we review the clinical and experimental evidence that supports the role of TH in neurodegenerative diseases.

## 2. Role of THs in Dementia

Cognitive decline is a common feature of overt hypothyroidism and ordinarily can be reversed with levothyroxine therapy. Cognitive decline is also documented in clinical and subclinical hyperthyroidism [7,8]. However, in people over the age of 75 years, subclinical hypothyroidism does not have significant clinical sequelae [9,10,11,12] except possibly depression [13], and in those aged 85 years or older, mild TSH elevation is associated with longevity [11,12], while higher serum levels of T4 are associated with increased mortality [11].

A cross-sectional study of 335 study volunteers aged 75 years and older did not find an association between elevated TSH concentrations up to 10 mIU/L and cognitive decline [10]. Similarly, in another study of 559 individuals aged 85 to 89 years, hypothyroidism was not associated with impaired cognitive function [11], although women with modestly elevated TSH concentrations might be at higher risk of cognitive decline [5]. Thus, individuals older than 75 years with TSH levels lower than 10 mIU/L do not have significantly reduced cognitive function [9,14,15,16]. In contrast, mild thyrotoxicosis is associated with increased risk of dementia [17,18,19].

In a longitudinal study of 313 euthyroid participants with a mean age of 72.5 years, the risk of progression of cognitive dysfunction over 5 years was 1.7 times higher for every 1 mIU/L decrease in baseline serum TSH. [18]. In another study of patients on suppressive treatment with levothyroxine for post-ablative management of thyroid cancer, subclinical hyperthyroidism was associated with significant declines in executive functions, psychomotor speed, and attention [19]. These observations were not supported in a subsequent study using relatively less sensitive testing with Mini-Mental State Examination (MMSE) [20]. It is noteworthy that in an interventional study, methimazole treatment to reduce TH production prevented declines in MMSE scores, while the untreated subjects with lower TSH levels had significant reductions in MMSE scores [21]. Thus, modestly increased serum TH levels may adversely affect cognitive capacity.

## 3. Role of TH in PD

The association between PD and TH regulation and dysregulation has been observed in several studies [22,23,24]. Parkinson’s disease is associated with altered TH economy, and the treatment of PD with levodopa or bromocriptine can interfere with the hypothalamic–pituitary–thyroid axis. Thyroid disorders, including hypothyroidism, Hashimoto’s thyroiditis, hyperthyroidism, and Graves’ disease, increase the risk of PD but also share some clinical signs with PD [25]. In addition, there is a significant association between TH levels and the degree of cognitive impairment in PD patients [26].

The ras homolog enriched in the striatum (*Rhes*) gene is induced by TH and expressed in many areas of the adult brain. It modulates dopamine (DA) neurotransmission by inhibiting the striatal cAMP/protein kinase A pathway [27]. The Rhes protein modulates L-DOPA-induced dyskinesia in PD rodent models and Rhes is involved in the survival of mouse midbrain dopaminergic neurons of the substantia nigra pars compacta (SNc), thereby contributing to mechanisms underlying dysfunctions of the nigrostriatal system [27]. Pharmacological dopamine depletion reduces Rhes expression in animals and Parkinson’s patients, indicating that there is a link between Rhes and PD.

Thyroid hormones and some of their derivatives induce differentiation of rat neural precursor cells (NPCs) into DA neurons through the nuclear receptor-related 1 (Nurr1) protein, an orphan nuclear receptor [28]. Both T3 and T4 are equally efficacious at inducing DA neuron differentiation; however, they are not useful as therapeutic agents due to their effects on the hypothalamic–pituitary–thyroid axis. However, two derivatives, methyl (S)-2-acetamido-3-(4-(4-acetoxy-3,5-diiodophenoxy)-3,5-diiodophenyl)propanoate and (S)-2-amino-3-(4-(4-hydroxy-3,5-diiodophenoxy)-3,5-diiodophenyl)-N-((1R,2S)-2-phenylcyclopropyl) propanamide hydrochloride, also induce DA neural differentiation but have no effect on the expression of a TH-responsive reporter gene. They also protect DA neurons from neurotoxic insults. It remains to be seen if these or other TH analogs have a role in the treatment of PD.

Thyroid hormones continue to regulate neurogliogenic processes in the mature mammalian brain and control the generation of new neuronal and glial progenitors from neural stem cells. They are also involved in adapting adult mice both metabolically and behaviorally to environmental changes by coordinating transcriptional programs in the frontal cortex [29]. Thus, PD symptomology may be in part mediated by altered TH function in cortical neurons. It remains to be seen if THs could help to restore neuronal and oligodendrocytes loss in some neurodegenerative diseases [30].

## 4. Pathophysiology of TH-Related Neurodegeneration

### 4.1. Cerebral Vasculature and TH

The BBB is endowed with various transport proteins that selectively transport nutrients and electrolytes while preventing the free crossing of solutes into the CNS. The TH transporters monocarboxylate transporter 8 (Mct8) and organic anion-transporting protein 1c1 (Oatp1c1) are expressed in the BBB and mediate the transfer of TH from the systemic circulation to the CNS [31]. The TH analogs sobetirome (GC1), Tetrac, and 3,5-diiodothyropropionic acid (DITPA) cross the BBB in sufficient quantities to modulate TH-responsive gene expression and neuronal cell stress or differentiation [32,33,34].

Aging alters the BBB transport of several solutes, suggesting a deterioration of barrier function. One of the age-related changes in the BBB in rats is decreased transport of levo-T3, while the transport of dextro-T3 is not altered [35]. This stereo-selective transport is counter-balanced with reduced clearance of the T3 content of cerebrum in aged rats [35]. Hypothyroidism in rats is associated with reduced BBB transport of β-hydroxybutyrate, while the BBB transport of hexoses, neutral amino acids, basic amino acids, and monocarboxylic acids are not altered [36].

The well-known effects of TH on cardiovascular disease extends to changes in cerebral vasculature. In hypothyroid patients, the risk of strokes is increased in those aged 18–64 years old [37] but not in patients 65 years old and over [37,38]. In addition, decreased cardiac output in the hypothyroid state reduces cerebral perfusion, thereby contributing to neurocognitive disorders. The extent of the contribution of thyroid dysfunction-related changes in the cardiovascular system to neurodegenerative changes in aging is not known.

### 4.2. Role of TH in Neuronal Cell Viability

In animal models, TH has been found to have a protective role in traumatic brain injury (TBI) [39,40,41], as well as in hypoxic brain injury [42,43,44]. These studies highlight the salutary effects of TH in neuronal survival. However, TH excess can cause cell death in primary neuronal cultures and in PC12 cells (Figure 1 and Figure 2) [45,46]. This neurotoxicity is at least in part mediated by the TH-responsive protein (THRP) (NM_003251) [45,46,47,48]. THRP is 83% homologous to the mouse c-Abl-interacting protein-2 (Abi2), a c-Abl-modulating protein with tumor suppressor activity [47,48]. Abi-2 is expressed in mid-to-late-stage embryos and in postnatal cortical neurons [49]. Furthermore, Abi-2 deficiency in mice leads to defects in neuron migration, aberrant dendritic spine formation, and learning and memory defects [50]. In neuronal cell cultures, T3 increases the expression of THRP and causes cell death. Exogenous THRP expression in primary neuronal cell cultures causes cell death (Figure 3). In addition, exogenously expressed THRP, through transfection, causes PC12 cell death (Figure 4 and Figure 5) [45,46]. However, the expression of exogenous THRP in other cells such as the colonic epithelial cell line Caco-2 and the glial cell line U251 has no effect on cell viability [46]. In PC12 cells, cell death occurs primarily by necrosis (Figure 4 (note lack of chromatin condensation with Hoechst 33342) and Figure 5 (intense signal with propidium iodide)), although cell cycle arrest may also occur [46]. Importantly, the 5′-flanking region of THRP has a putative TH-response element (TRE) that is conserved in rats and humans [51]. This effect of THRP on cell viability is not modulated by c-Abl tyrosine kinase [46].

Another TH responsive gene in adult rat cerebral tissue is the novel translational repressor N-acetyltransferase-1 (Nat-1) (NM_001160170) [52]. Nat-1 mRNA is widely expressed in various tissues, and in hepatic tissue, it is also TH-responsive. In aging rats, there is a significant reduction in the TH responsiveness of THRP and Nat-1 [52,53]. The latter is consistent with the literature on age-related resistance to TH action [54].

Thyroid hormone also has neuroprotective effects in animals and humans that is best demonstrated by the rapid response to traumatic brain injury (TBI). Neurons and glial cells express the TH transporters Mct8 and Oatp1C1 [55,56]. In the mouse brain, Mct8 and Oatp1C1 are expressed in temporal and spatially unique patterns regulating T4 uptake in a precise manner [55,56]. Once transported into the cell, T4 is converted to T3 by the T4 5′-deiodinase 1 (Dio1) and 5′-deiodinase 2 (Dio2) (in rodents), which are expressed in both neurons and glial cells [57,58]. 5′-deiodinase 3 (Dio3), which catalyzes the conversion of T4 to rT3, protects the cortex, cerebellum, and sensory neurons from premature T3 exposure [59]. In developing neurons, differentiation factor sonic hedgehog induces neuron proliferation, increases Dio3 expression, and suppresses Dio2 via ubiquitination and proteasome-mediated proteolysis [60]. Thus, TH governs many critical aspects of neuronal regulation. Disturbances in any of these factors has the potential to alter neurological function. This is demonstrated in Allan–Herndon–Dudley Syndrome patients in which MCT8 mutations lead to reduced TH activity, delayed myelination by oligodendrocytes, and severe X-linked mental retardation [61,62].

In mouse models, TBI elicits neuronal damage that is repaired in distinct stages [39,40,41]. Patients with TBI have reduced serum T4 and T3 levels and a normal TSH, which is referred to as Non-Thyroidal Illness [63], though some may have central hypothyroidism if the pituitary is involved [64]. Diffuse TBI models are similar to brain concussions with axonal injury, ischemia, vascular damage, and brain edema. In contrast, focal TBI models simulate loss of local blood flow to a specific area of the brain. Both models reduce blood supply, promote brain swelling, increase neuronal cell death, and cause learning, memory, and cognition loss.

Controlled cortical injury (CCI) is frequently used to simulate focal injury in mouse TBI [65]. Depending on the depth of contusion, different degrees of injury can be modeled. Several studies have shown that CCI-induced lesions can be reversed by post-TBI TH administration. In rats, TH administration (2.5 g/100 g) i.p. 1 h post-TBI significantly reduced edema [40]. In a mouse model, TH administration (1.2 μg/kg) 1 h post-TBI also reduced brain edema [41], while in a stroke model (either transient or permanent middle cerebral artery occlusion), one TH dose (25 μg/kg) either pre- or post-stroke reduced both ischemia and edema [44]. The reduction in brain edema may be due to TH-mediated aquaporin 4 inhibition [44]. Improvements in ischemia post-TH treatment may be related to elevated NO since T4 has been shown to induce inducible nitric oxidase synthase (iNOS) synthesis in postnatal cortical neurons [44]. Though T4 administration had no effect on hypoxia-inducible factor 1α (HIF1α) and nuclear factor-κB (NF-κB) expression, iNOS mRNA increased 3.4-fold [66]. Treatment with TH pot-TBI also reduced pro-apoptotic-related genes such as BCL2 [67] and neurotrophin production [68]. These observations suggest that TH also reversed neuronal cell stress to preserve brain function, though in a transient manner.

Thyroid hormone can alter neuronal cell viability indirectly by augmenting adrenergic neurotransmission. Thyroid hormones increase β-adrenergic receptor number and increase the responsiveness of β-adrenergic receptor activity in synaptosomal membranes and cerebral microvessels [69,70]. The increased adrenergic tone may contribute to increased cellular stress.

### 4.3. Role of TH in Cellular Stress

Thyroid hormones may have a role in the pathogenesis of Alzheimer’s disease through cellular stress, notably oxidative and inflammatory stress [71,72,73]. TH treatment of rats increases peroxidation byproducts such as ethane exhalation in vivo [74]. However, daily T3 treatment of rats does not alter the cerebral tissue content of malondialdehyde (MDA)-modified proteins [75]. Increased ATP synthesis in response to the metabolic effects of TH treatment enhances reactive oxygen species (ROS) production. In the absence of adequate antioxidative functions to counter-balance these effects, excess TH may promote neurodegeneration. However, as demonstrated in TBI, TH treatment inhibits transient oxidative stress and ROS production [39,40,41], inhibits apoptosis [76], and enhances Ca^2+^ transport to the sarcoplasmic reticulum (in muscle cells) and endoplasmic reticulum (all cells) via sarcoendoplasmic reticulum calcium ATPase induction [77]. The biological and clinical implications of these changes are still unknown. It is tempting to speculate that the long-term and persistent increased cellular stress induced by THs may promote aging, PD, and AD.

### 4.4. TH Effect on β-Amyloid Precursor Protein

Thyroid hormones may have a role in the pathogenesis of AD by altering the splicing of the β-amyloid precursor protein (APP) gene that yields three major APP messenger RNAs (mRNAs), which, in turn, give rise to the APP770, APP751, and APP695 protein isoforms [78]. In N2a-beta neuroblastoma cells, T3 treatment increased the amount of APP770 and reduced the immature APP695 isoform. Indeed, T3 regulates not only APP gene splicing but also the processing and secretion of the APP peptides [78]. The changes in the intracellular and extracellular contents of APP isoforms may promote Alzheimer’s disease.

In support of this hypothesis, a recent systematic review and meta-analysis examining the relationships between thyroid hormone status and prevalence of AD suggested that serum TT3 and FT3 and cerebral spinal fluid (CSF) TT3 are lower in AD patients [79]. In 32 studies, there were no significant differences in serum TSH, TT4, and FT4 in the AD group compared to the controls. There were also no significant differences in CSF TT4 and TT3/TT4 ratios [79]. Although these observations should be examined in a longitudinal study, these results and those described above suggest that reductions in serum total and free T3 and CSF total T3 may be a protective response.

## 5. Conclusions

There are a number of biochemical and physiological effects of THs on the brain throughout the human lifespan. Prenatal TH effects are essential for embryonic development, though many of these effects are transient. During the growth and reproductive phases, THs are essential for behavior and metabolic adaptation to the environment. However, nutrient excess increases hypothalamic–pituitary–TH axis activity, glucose and lipid metabolism, mitochondrial ROS generation/antioxidant depletion, and inflammatory cytokine release. Importantly, T3 and T4 also induce cell death in neurons (Figure 1 and Figure 3) suggesting that persistent hyperthyroidism may promote neurodegenerative changes. To counter this, T3, T4, and TH transport decreases; Dio1 and Dio2 activity decreases; and Dio3 activity increases in the elderly. Thus, TH plays an important role in the survival and differentiation of neurons and may be implicated in the pathogenesis of neurodegenerative diseases, such as Alzheimer’s disease and Parkinson’s disease [80].

Prior studies examining TH-responsive genes in the brain using subtractive hybridization, differential display, and microarrays have many limitations. Current methods such as RNAseq and single-cell-RNAseq profiling [29] have the potential to drive further advances in the field at a much faster pace. These scientific revelations may help to identify potential preventive measures to reduce the risk of neurodegenerative diseases. 

## Figures and Tables

**Figure 1 cells-14-00140-f001:**
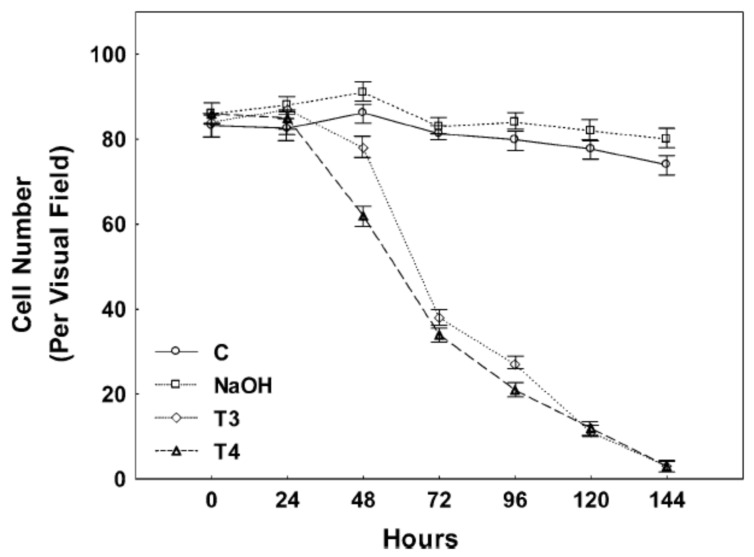
Neuronal cells were maintained for 24 h in Neurobasal medium and treated with vehicle (50 mM NaOH), 100 nM T3, or 100 nM T4. The media for control and treated cells were changed and thyroid hormone replenished daily for six consecutive days. The number of cells per 1 mm^2^ was determined with an inverted microscope with phase-contrast illumination. Experiments were performed in triplicate, and three regions in each well were counted. The number of cells was counted each day for six consecutive days of treatment. The experiment was repeated three times. Neuronal viability was significantly reduced in T3-treated cultures by 48 h, while the changes seen following T4 treatment became significant at 72 h of incubation. Few neurons survived after six days of either T3 or T4 treatment. Untreated control and vehicle-treated cells continued to survive until the end of the experiment. The figure is used with permission from Springer Nature, Cham, Switzerland [45].

**Figure 2 cells-14-00140-f002:**
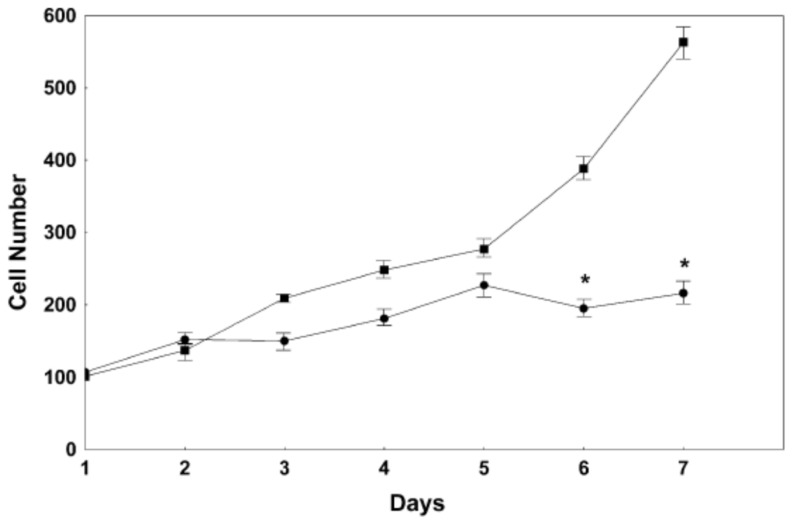
The effect of T3 on PC12 cell proliferation. The cells were treated with 100 nM T3 (●) or solvent (50 mM NaOH) (■), and the cell number was measured on day one and for six consecutive days. Treatment with T3 decreased PC12 cells significantly on days 6 and 7. N = 6, *, *p* < 0.001 relative to control cells. The figure is used with permission from Springer Nature [46].

**Figure 3 cells-14-00140-f003:**
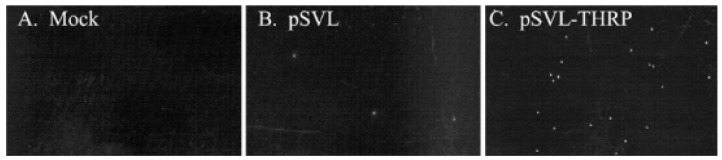
The effect of exogenous THRP expression in primary neuronal cell cultures on propidium iodide (PI) staining. Primary neuronal cell cultures were transfected with no DNA (**A**) or with pSVL (**B**) or pSVL-THRP (**C**). After 48 h, cells were stained with PI, counted, and photographed. Few PI staining cells were observed in mock transfected cells and in cells transfected with the empty vector pSVL. However, PI-positive cells were abundant in cells transfected with the THRP expression plasmid. The figure is used with permission from Springer Nature [45].

**Figure 4 cells-14-00140-f004:**
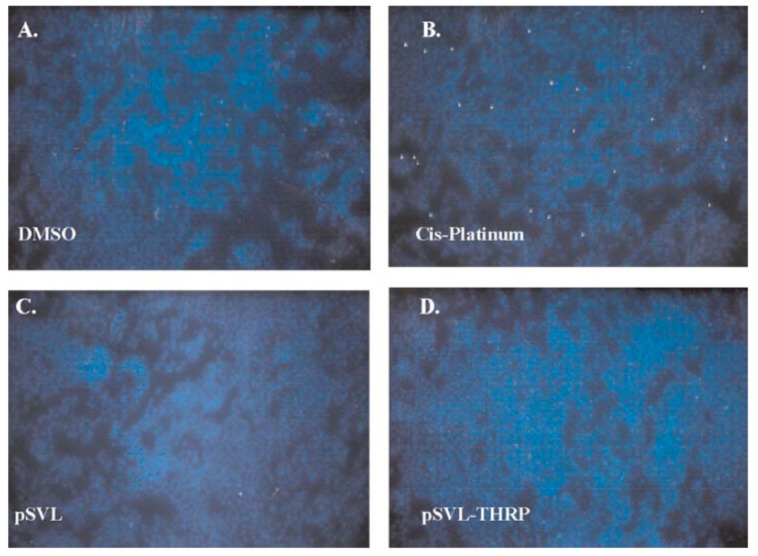
The effect of THRP on apoptosis in Hoechst 33342 (HOE)-stained PC12 cells. Cells were treated with the solvent dimethylsulfoxide (DMSO) (**A**) or cis-platinum dissolved in DMSO (**B**) for 24 h and stained with HOE to measure apoptosis-induced chromatic condensation. Cells were transfected with the empty vector pSVL (**C**) or the THRP expression plasmid pSVL-THRP (**D**) and stained with HOE after 48 h. Treatment with cis-platinum increased the number of HOE-positive cells, but no positive cells were observed in cultures transfected with the empty vector and the THRP expression plasmid was negative. The figure is used with permission from Springer Nature [46].

**Figure 5 cells-14-00140-f005:**
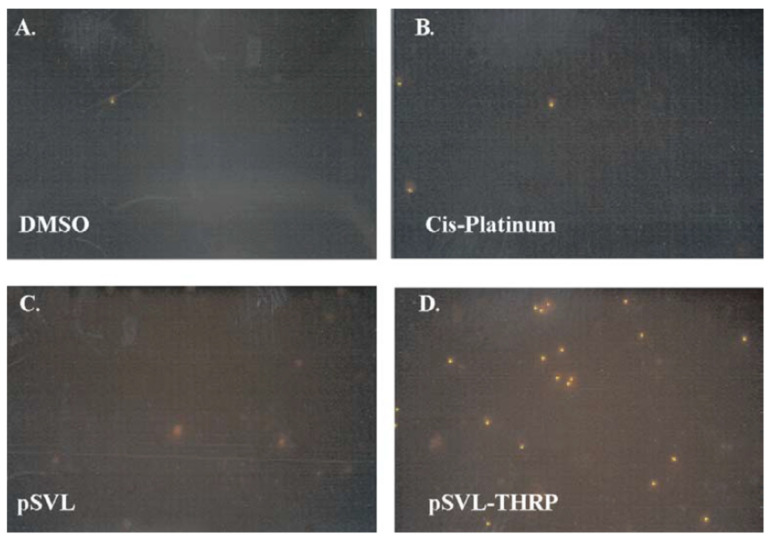
The effect of THRP on propidium iodide (PI) staining in PC12 cells. Cells were treated with the solvent dimethylsulfoxide (DMSO) (**A**) or cis-platinum dissolved in DMSO (**B**) for 24 h, and cells were stained with PI. Cells were transfected with the empty vector pSVL (**C**) or the THRP expression plasmid pSVL-THRP (**D**) and 48 h later stained with PI. PI-positive cells increased slightly in cells treated with cis-platinum but were numerous in cells transfected with the THRP expression plasmid relative to the empty vector. The figure is used with permission from Springer Nature [46].

## Data Availability

Not applicable.

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
