# Peer review of "Role of Thyroid Hormone in Neurodegenerative Disorders of Older People"

_cells, 2025, doi:10.3390/cells14020140_

Round 1
Reviewer 1 Report
Comments and Suggestions for Authors
In this review, the authors have reviewed clinical and experimental evidences to explore the potential role(s) of thyroid hormone in the aetiology/pathology of neurodegenerative diseases.
Overall the review was well written- the structure was logical with clear background information. In addition, the authors have comprehensively the various aspects of thyroid hormones in neurodegenerative diseases in terms of disease risk and the underlying molecular mechanisms. Just a few suggestion:
1) For the section " Role of TH in dementia" (starting from Line 38), is it possible to tabulate the main findings from the clinical studies
2) For the section "Role of TH in PD" (starting from Line 69), is it possible include a few clinical studies that showed direct relationship between thyroid hormone and disease risk. Indeed, there are a few such reports such as Xu, J (2022) Clinical and Experimental Immunology; Chen, SF (2020) Parkinsonism and Related Disorders; Charoenngam, N (2022) Frontiers in Endocrinology to a name a few.
3) Figure 3: Figure legend is missing.
4) It would be interesting to discuss/postulate how these knowledge can translate into potential treatment/ therapeutics for neurodegenerative diseases especially in terms of the molecular mechanisms described in the review
Author Response
For the section " Role of TH in dementia" (starting from Line 38), is it possible to tabulate the main findings from the clinical studies
We elected to forgo the description of the details of neuropsychiatric testing results to avoid distraction from the main message in the manuscript.
2) For the section "Role of TH in PD" (starting from Line 69), is it possible include a few clinical studies that showed direct relationship between thyroid hormone and disease risk. Indeed, there are a few such reports such as Xu, J (2022) Clinical and Experimental Immunology; Chen, SF (2020) Parkinsonism and Related Disorders; Charoenngam, N (2022) Frontiers in Endocrinology to a name a few.
The important studies the Reviewer mentions that have shown the relationship between thyroid hormones and Parkinson’s Disease risk is now included in the revised manuscript.
3) Figure 3: Figure legend is missing.
The legend is now corrected.
4) It would be interesting to discuss/postulate how these knowledge can translate into potential treatment/ therapeutics for neurodegenerative diseases especially in terms of the molecular mechanisms described in the review.
We now have added a statement in the Conclusion section as to the potential therapeutic potential of manipulating thyroid hormone effects in the CNS to reduce neurodegenerative disease.
Reviewer 2 Report
Comments and Suggestions for Authors
The authors conducted a non-systematic review on the role of thyroid hormones in neurodegeneration pathology, in particular Parkinson's and Alzheimer's disease. The type of review is not explicitly stated in the title. The article is divided into 5 sections including introduction and conclusions, with the most extensive discussion in the “Pathophysiology of TH-Related Neurodegeneration” section. The article achieves the goal stated in the title well, that is, to highlight the role of thyroid hormones in neurodegeneration, with clear and logical writing that helps the reader to orient themselves in the wealth of information. The cited literature is sufficient to support the article's descriptive and interpretative parts. The conclusions are supported by the reported data
In my opinion, the article could be published as it is, with only two marginal notes:
Line 28: Thyroid hormones (TH) affect neurotransmission, cause oxidative and inflammatory stress ….. not thyroid hormones' physiological role but their abnormal blood level
Figure 3 does not show the graphic identification of the two lines with different marks
Author Response
Line 28: Thyroid hormones (TH) affect neurotransmission, cause oxidative and inflammatory stress ….. not thyroid hormones' physiological role but their abnormal blood level.
We corrected the sentence as was recommended by the Reviewer. Thank you for the suggestion.
Figure 3 does not show the graphic identification of the two lines with different marks
The Figure 3 legend is now corrected.